# UniLP: Unified Topology-aware Generative Framework for Link Prediction in Knowledge Graph

## ABSTRACT

Link prediction (LP) in knowledge graph (KG) is a crucial task that has received increasing attention recently. Due to the heterogeneous structures of KGs, various application scenarios, and demand-specific downstream objectives, there exist multiple subtasks in LP. Most studies only focus on designing a dedicated architecture for a specific subtask, which results in various complicated LP models. The isolated architectures and chaotic situations make it significant to construct a unified model that can handle multiple LP subtasks simultaneously. However, unifying all subtasks in LP presents numerous challenges, including unified input forms, task-specific context modeling, and topological information encoding. To address these challenges, we propose a topology-aware generative framework, namely UniLP, which utilizes a generative pre-trained language model to accomplish different LP subtasks universally. Specifically, we introduce a context demonstration template to convert task-specific context into a unified generative formulation. Based on the unified formulation, to address the limitation of transformer architecture that may overlook important structural signals in KGs, we design novel topology-aware soft prompts to deeply couple topology and text information in a contextualized manner. Extensive experiment results demonstrate that our framework achieves substantial performance gain and provides a real unified end-to-end solution for the whole LP subtasks. We also perform comprehensive ablation studies to support in-depth analysis of each component in UniLP. The code is available at https://anonymous.4open.science/r/UniLP/.

## CCS CONCEPTS

• **Do Not Use This Code → Generate the Correct Terms for Your Paper**; *Generate the Correct Terms for Your Paper*.

## KEYWORDS

Link Prediction; Unified Generative Framework; Soft Prompt; Knowledge Graph

**ACM Reference Format:**
Anonymous Author(s). 2018. UniLP: Unified Topology-aware Generative Framework for Link Prediction in Knowledge Graph. In *Proceedings of Make sure to enter the correct conference title from your rights confirmation emai (Conference acronym 'XX).* ACM, New York, NY, USA, 11 pages. https://doi.org/XXXXXXX.XXXXXXX

## 1 INTRODUCTION

Human-curated knowledge graphs (KGs) provide critical supportive information to various semantic web technologies, but these graphs are usually incomplete, urging auto-completion of them. Therefore, link prediction (LP) in KGs aims to infer missing or future facts, which has received extensive attention in recent years. Due to the variety of KG structures and different downstream task objectives, LP derives a plethora of subtasks, including static link prediction [1, 51], inductive link prediction [24, 36], few-shot link prediction [6, 54], and temporal link prediction [15, 17].

**(a) Various LP subtasks**

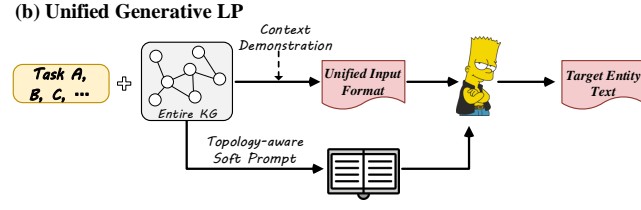

**(b) Unified Generative LP**

**Figure 1: (a) Various LP subtasks: heterogeneous context structures and isolated model architectures. (b) Our topology-aware unified LP: universally modeling via generative PLMs with prompt learning.**

Currently, most LP approaches are task-specialized, which leads to dedicated architectures, isolated models, and specialized knowledge sources for different LP subtasks. And the corresponding modeling advances make the progress of each subtask seemingly unique and incompatible. Fig 1 (a) presents illustrations of all subtasks with specific examples. Different types of KG, such as static KG or temporal KG, lead to highly specialized encoders [17, 47]. Moreover, to

adapt to more realistic and challenging downstream task objectives, e.g., few-shot setting and inductive setting, the additional training scheme (e.g., meta-learning) is applied in some subtasks [7, 26]. Therefore, instead of solving common challenges in LP research, improvements in LP have been prone to design a new architecture carefully and trained with the corresponding dataset for a specific subtask. The isolated architectures and chaotic situations make it significant to construct a unified model that can handle multiple LP subtasks simultaneously without relying on the sub-models or changing the model architecture.

Recently, with powerful knowledge sharing and semantic generalization capabilities of pre-trained language models (PLMs), many natural language processing (NLP) tasks (e.g., name entity recognition [22], aspect-based sentiment analysis [12, 50]) have been modeled into a unified framework. A straightforward idea is to unify the LP task by transforming all subtasks into text-to-text formats and then generate target text via large-scale generative PLMs [28]. Some methods [4, 44] instantiate this idea by utilizing entity description texts and yield encouraging results. In this way, on the one hand, the flexibility of text makes it possible to model multiple subtasks universally, and on the other hand, the autoregressive decoding in generative PLMs can significantly improve inference efficiency compared to conventional LP approaches scoring all entities. Despite their success, these methods suffer from their inherent problems, which limit their potential and performance in universal modeling. Firstly, they rely heavily on the quality of entity description texts but fail to model the contextual knowledge (e.g., entity local structure, temporal snapshots) uniformly in KGs. Secondly, the structure topological information, a crucial aspect of KGs, is inevitably lost when simply linearizing KGs into text form.

The issues above prompt us to rethink the unification of LP tasks. Fundamentally, the unified modeling all subtasks of LP presents three primary challenges: (i) **Input Format**. The input formats of each subtask vary. For example, in Figure 1 (a), the input query is a simple triple format in static and inductive settings, while it takes on a quadruple form in the temporal setting. In contrast, in the few-shot setting, there is a set of support edges associated with query relation in addition to the query triple. Unifying the input format of all subtasks is the first challenge. (ii) **Context Modeling**. Different subtasks aim for diverse context modeling. Some subtasks use entity local context (e.g., static LP, few-shot LP), while others use transferable entity-independent relational context (e.g., inductive LP). The formats and representations for different contexts differ, and each context expresses information in its unique way. For example, entity-related context uses a local subgraph, relational context uses an enclosing subgraph, and temporal context uses a series of subgraph snapshots. Representing diverse contexts over graphs with a unified representation is the second challenge. (iii) **Topological Encoding**. Employing the transformer architecture [40] often overwhelmingly focuses on the textual information and overlooks structural knowledge, which is a loss to the text encoder because network signals are often strong indicators of text semantics. Although some methods [41, 42] adopt Siamese-style PLM encoder to learn structure and text information, they are task-specialized and employ encoder-only architecture. Therefore, the third challenge is how to deeply couple topology and text information in a contextualized way.

In response to the challenges of universal LP, we propose UniLP, a unified topology-aware generative framework that can universally address all the LP subtasks. First, we transform all subtasks into a generative paradigm with an encoder-decoder T5 [28], which could handle the obstacles on the input and KG types without any model architecture changes. Then, to model heterogeneous graph context structures, we design a context demonstration template that can effectively encode different context structures into a uniform representation so that various subtasks can be universally modeled in the same text-to-text generation framework. Finally, we design a topology-aware soft prompt to simultaneously model the deep interactions between topology and text information and establish the correspondence between KG elements and the prompt vectors. Unlike previous methods [19] that prepend soft prompts to the inputs of PLMs, we treat prompt and input text as different elements by interpolating the soft prompts into inputs. Such design effectively avoids over-fitting towards textual information and fuses the topological and textual information. To demonstrate the effectiveness of our proposed framework, we conduct extensive experiments on several challenging benchmarks of all LP subtasks. Comparison results show that our proposed UniLP outperforms most state-of-the-art (SOTA) models in every subtask.

In summary, our main contributions are as follows:

- We propose a novel prompt-based approach to link prediction, UniLP, which unifies all subtasks under a single generative framework. By introducing a context demonstration template that encodes various context structures into a uniform representation, UniLP achieves a unified solution without the need for dedicated architectures or extensive changes in the model structure.
- We innovatively propose topology-aware soft prompts to deeply couple topology and text information in a contextualized manner, addressing the limitation of transformer architecture that may overlook important structural signals in KGs.
- We conduct extensive experiments on all subtasks, and the experimental results show that our framework consistently improves over the baselines. To the best of our knowledge, it is the first work to evaluate a model on all LP subtasks.

## 2 RELATED WORK

***Structure Learning for Link Prediction.*** Recent advancements in various subfields of LP mainly explore structure knowledge in KGs through spatial measurement or latent matching. Specifically, spatial measurement models intensively use translation-based scoring functions to measure the distance between two entities. TransE [1] defines each relation as a translation from the subject to the object. RotatE [34] further extends TransE to model symmetry relation patterns. Semantic matching methods, such as ComplEx [38] and DistMult [51], calculate the semantic similarity of representation. Moreover, due to the natural graph structure of KGs, some graph neural networks (GNN) based methods [31, 39] achieve promising results. Following static LP methods, TTtransE [15] and HyTE [9] encode time in the entity-relation dimensional spaces with time embeddings and temporal hyperplanes. Similarly, MetaR [6] and

GMatching [45] exploit meta-learning to address the issue of long-tail relations in KGs. GraIL [36] and COMPILE [24] extracts the enclosing subgraph and use double radius labeling to address the challenges in the inductive setting. Despite their success, separate solutions for standalone subtasks hinder the development of LP research, and it may be costly to deploy a variety of specialized models in practice.

**PLM-based Methods for Link Prediction.** Pre-trained language models, including BERT [11], GPT [3], and T5 [28], have shown effectiveness in capturing general language representations and have led to a learning paradigm shift in NLP. Inspired by this, KG-BERT [53], StAR [41], and SimKGC [42] convert LP into a sequence classification problem with the binary cross-entropy object. In the temporal LP subtask, PPT [48] utilizes the PLMs to capture the implicit temporal correlations among relations in KGs. However, these encoder-only methods costly scoring of all possible triples in inference and suffer from unstable negative sampling. Instead of calculating scores from embeddings, KGT5 [30] and KG-S2S [4] explore the sequence-to-sequence (Seq2Seq) PLM models to generate target entity text on LP tasks directly. GenKGC [44] further proposes an entity-aware hierarchical decoding strategy for fast inference. Compared with existing Seq2Seq-based LP approaches, the most significant difference is that previous approaches rely heavily on text information and ignore the structure topology information of KGs. We aim to deeply couple the topology and text information in a contextualized way. Besides, we focus on training a versatile model that handles all subtasks in LP without changing the model structure.

**Prompt Learning.** To fill the gap between the objective of PLMs and the downstream fine-tuning objective, prompt learning is proposed to leverage the implicit knowledge stored in PLMs by adding additional hints [3]. For example, Shin et al. [33] extend this paradigm and propose hard prompts described via natural language templates to improve the generalization ability of the model. To relax the constraint that templates are natural language tokens, Lester et al. [19] and Liu et al. [21] introduce additional trainable parameters as soft prompts achieve comparative performance on various NLP tasks, where each soft prompt corresponds to a learnable embedding. In this work, we utilize a discrete text prompt template to model the appropriate context for each LP subtask in a unified formulation. Then, we propose topology-aware soft prompts to inject graph topological signals into the PLMs encoding process.

## 3 METHODOLOGY

### 3.1 Link Prediction Task Unification

In this paper, we aim to solve static, inductive, few-shot, and temporal LP in a unified framework. For a detailed description of each task, refer to Appendix A.1. Without loss of generally, a KG is defined as $\mathcal{G} = (\mathcal{V}, \mathcal{R}, \mathcal{E}, \mathcal{M})$, where $\mathcal{V}, \mathcal{R}, \mathcal{E}$, and $\mathcal{M}$ represent the sets of entities, relations, edges and meta information, respectively. Each edge $e \in \mathcal{E}$ is a quadruple $(m, s, r, o) \in \mathcal{M} \times \mathcal{V} \times \mathcal{R} \times \mathcal{V}$ which connects head entity $s$ and target entity $o$ with relation type $r$, associated with meta information $m$. The meta information $m$ denotes different forms of contents in different link prediction settings, which could be null, i.e., $\mathcal{M} = \emptyset$ in the static and inductive

settings, $K-$shot support edges of a relation for few-shot setting, or timestamp in temporal setting. To effectively perform link prediction across diverse settings using a unified model architecture, we formulate the four link prediction tasks as a Seq2Seq problem, i.e., directly generating target entities for the given query. Formally, given a query $q = (m, s, r, ?)$ or $q = (m, ?, r, o)$, when combined with a text prompt, scoring the target entity in different link prediction subtasks can be uniformly formulated as a Seq2Seq problem,

$$score(v_j|q) = p(Y|q) = \prod_{i=1}^{m} p(y_i|Y_{<i}, q), \quad (1)$$

where $v_j \in \mathcal{V}$ is the target entity, $Y_{<i}$ represents the prefix of target entity sequence $Y$ up to position $i-1$, and $p(y_i|Y_{<i}, q)$ represents the probability of generating token $y_i$ given $Y_{<i}$ and $q$.

### 3.2 Overview of the approach

Based on the above task formulation, we develop a novel unified framework to serve a variety of link prediction subtasks. To implement such a unified approach, we need to address two major challenges: (i) how to verbalize the KG context and query across diverse task settings into a single generative form (Section 3.3), and (ii) how to properly inject topological information in KGs into the text encoding process in a contextualized way (Section 3.4). Additionally, we introduce the training and inference process (Section 3.5). The overall architecture of our proposed model is illustrated in Figure 2.

### 3.3 Context Demonstration Template

In this section, we focus on (i) modeling appropriate context $C_q$ for a given query $q$, (ii) converting $\{C_q, q\}$ into a unified prompt $\theta_q$. Prior works [39] have demonstrated the immense value of incorporating structure information when conducting link prediction. Nevertheless, the limited input length of PLMs restricts their ability to process the entire KG. Instead, we note that the relevant neighboring contexts around the query entity can serve useful signals to guide the generation process. For this purpose, given the query quadruple $q = (m, s, r, ?)$, we propose two context modeling templates by constructing $n$ demonstrations:

- **Entity-centered Demonstrations.** We sample demonstrations with the guidance of query entity $s$, which consists of one-hop neighbors related to the query entity $s$ from the training set. With a minor change, in the temporal link prediction setting, we sample historical facts related to entity $s$ based on time intervals and filter facts in current timestamp $m$, i.e., only retain facts from snapshots prior to timestamp $m$ for history context modeling. This ensures the contextual demonstrations come from the relevant historical time periods rather than the current state.

- **Relation-centered Demonstrations.** Different from entity-centered demonstrations focusing on local context, this template involves sampling facts that contain the same relation $r$. These demonstrations provide a global semantic perspective that facilitates the identification and characterization of relation $r$. It is worth noting that in the few-shot setting, we only adopt entity-centered demonstrations since the support edges of $r$ are already provided. Similarly, we

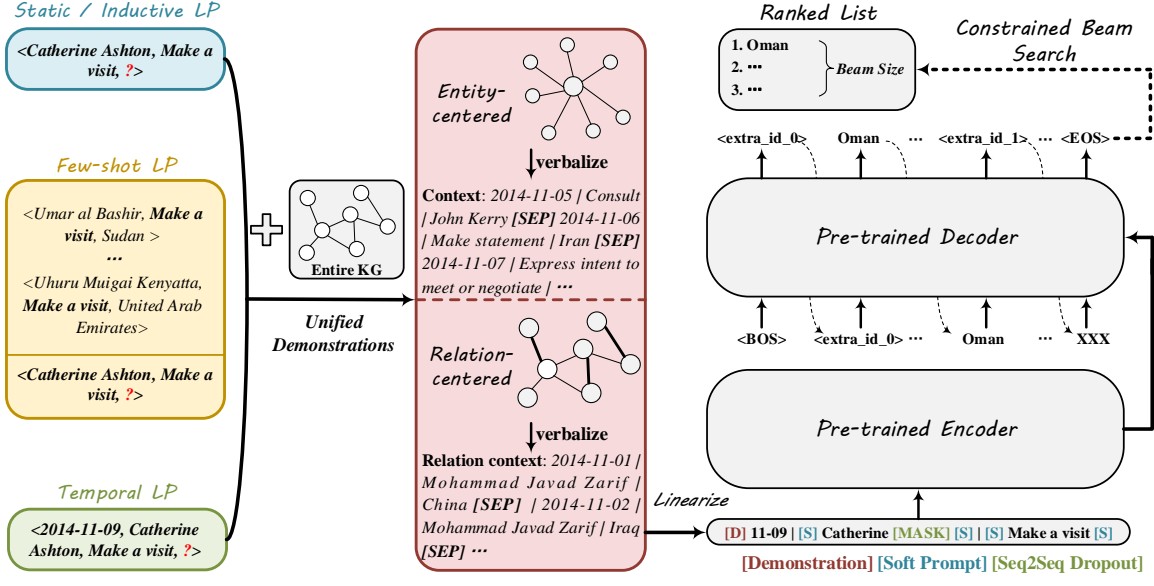

**Figure 2: Overview of UniLP: a novel approach for unified link prediction modeling. This shows an example of the temporal LP subtask for the query *<Catherine Ashton, Make a visit, ?, 2014-11-09>* from ICEWS14 [13]. We show two strategies to model the context of entity *Catherine Ashton* as demonstrations. In the training stage, we randomly mask input tokens to avoid over-fitting. In the inference stage, we adopt a constrained beam search algorithm to obtain top-$K$ valid entity mentions ($K$ is the beam size) as final answers.**

only retain facts prior to timestamp $m$ for history modeling under temporal LP.

Given the $n$ context demonstrations $C_q = \{c_1, c_2, \cdots, c_n\}$ and query $q$, we construct a prompt using a pre-defined template $\theta$, enabling unified formulation. Specifically, we verbalize each fact $c_i = (m_{c_i}, s_{c_i}, r_{c_i}, o_{c_i})$ in $C_q$ as $\langle m_{c_i} | \mathcal{N}(s_{c_i}) | \mathcal{N}(r_{c_i}) | \mathcal{N}(o_{c_i}) \rangle$ (e.g., *<2014-11-02 | Mohammad Javad Zarif | Make a visit | Iraq>*) where $\mathcal{N}(\cdot)$ is the lexical form of entity and relation. The query $q$ is then represented as $m | \mathcal{N}(s) | \mathcal{N}(r) | <Mask>$ where $<Mask>$ is the placing sentinel token of T5 pre-training, concatenated to the end of the context. We convert the template into a textual prompt $\theta_q$ by means of linearization [27]. More details are provided in Appendix A.6. Formally, for context demonstrations $C_q$ and a query $q$, we obtain the input sequence as:

$$X_q = [\langle BOS \rangle; c_1, c_2, \cdots, c_n; \langle SEP \rangle; \mathcal{N}(q)], \quad (2)$$

where $[;]$ denotes the sequence concatenation operation, $\langle BOS \rangle$ and $\langle SEP \rangle$ are separate markers in the applied PLMs. Moreover, we sample up to 20 demonstrations uniformly limited by the input length of PLMs. And we adopt Seq2Seq dropout by randomly masking $p\%$ of the input tokens in $X_q$ to address the over-fitting issue.

### 3.4 Topology-aware soft prompt

Linearizing a knowledge graph into text form inevitably results in the loss of structural information. While our proposed demonstration strategies have captured the local context around the query, this approach is somewhat limited, as they only model instance-level information but lack semantic-level information. To further

bridge the substantial gap between the link prediction task and the generative task, we propose topology-aware soft prompts to inject graph topological signals into the PLMs encoding process. The key idea is to introduce additional trainable prompt vectors for entities and relations, which capture global topological patterns in KGs and allow frequent interaction with the textual information in PLMs.

Specifically, given the query $q = (m, s, r, ?)$ and corresponding text prompt input $X_q$, let $\mathbf{H}_q^{(l)} \in \mathbb{R}^{d \times n}$ denote PLMs encoder output representations for all text tokens in $X_q$ after the $l$-th model layer ($l \geq 1$). As shown in Figure 3, We introduce four trainable prompt vectors $\mathbf{p}_{s_1}^{(l)}, \mathbf{p}_{s_2}^{(l)}, \mathbf{p}_{r_1}^{(l)}$, and $\mathbf{p}_{r_2}^{(l)} \in \mathbb{R}^d$, which are interpolated into the text token sequence hidden states as follows:

$$\widetilde{\mathbf{H}}_q^{(l)} = [\mathbf{H}_{<s}^{(l)}; \mathbf{p}_{s_1}^{(l)}; \mathbf{h}_s^{(l)}; \mathbf{p}_{s_2}^{(l)}; \mathbf{p}_{r_1}^{(l)}; \mathbf{h}_r^{(l)}; \mathbf{p}_{r_2}^{(l)}; \mathbf{H}_{>r}^{(l)}], \quad (3)$$

where $\mathbf{h}_s^{(l)}$ and $\mathbf{h}_r^{(l)}$ denote the $l$-th layer hidden states for the entity and relation tokens, $\mathbf{H}_{<s}^{(l)}$ and $\mathbf{H}_{>r}^{(l)}$ denote splits of $\mathbf{H}_q^{(l)}$ before entity and after relation, respectively. In this way, the distinct soft prompt vectors enable the decoupling of element-specific knowledge from the general textual knowledge in PLMs. Furthermore, to let soft prompt token representations carry structural signals, we propose a novel soft prompt representation method. Unlike previous methods [43] that train additional GNNs to encode structure information and directly integrate them into PLMs, our approach transforms the KG into a relational perspective, where each entity is characterized by its surrounding relations. Let $\mathbf{E}_{\mathcal{R}} \in \mathbb{R}^{|\mathcal{R}| \times d}$ denote the embeddings of relations in KG. Considering the distinct directions of relations characterize different semantics, we use a learnable relation-domain embedding matrix $\mathbf{E}_{\mathcal{R}}^{dom} \in \mathbb{R}^{|\mathcal{R}| \times d}$ and

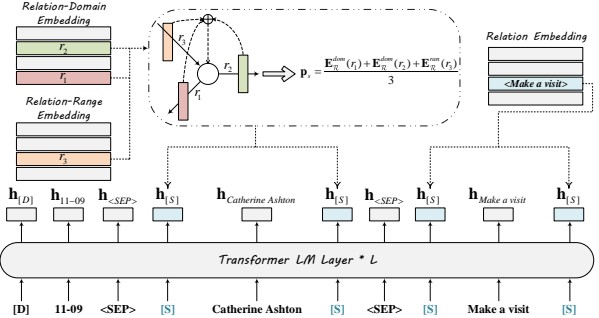

**Figure 3: The illustration of the topology-aware soft prompts initialization.**

a relation-range embedding matrix $\mathbf{E}_{\mathcal{R}}^{ran} \in \mathbb{R}^{|\mathcal{R}| \times d}$ to represent relational semantics of two directions. Formally, given the query $q$, we adopt the relation embedding to represent relation prompt representations, i.e., $\mathbf{p}_{r_1}, \mathbf{p}_{r_2} = \mathbf{E}_{\mathcal{R}}(r)$. The entity prompt representations are obtained by aggregating the surrounding relations of the query entity:

$$\mathbf{p}_{s_1}, \mathbf{p}_{s_2} = \frac{\sum_{r \in O(s)} \mathbf{E}_{\mathcal{R}}^{dom}(r) + \sum_{r \in I(s)} \mathbf{E}_{\mathcal{R}}^{ran}(r)}{|O(s)| + |I(s)|}, \quad (4)$$

where $I(s) = \{r | \exists x, (x, r, s) \in \mathcal{E}\}$ denotes the ingoing relation set for entity $s$, $O(s) = \{r | \exists x, (s, r, x) \in \mathcal{E}\}$ denotes the outgoing relation set for entity $s$. Figure 3 shows a visual illustration of the soft prompts initialization process. The resulting input embeddings $\widetilde{\mathbf{H}}_q^{(l)}$ carry inherent topology awareness, enabling the model to discern and leverage the underlying graph structure.

### 3.5 Training and inference

In our approach, link prediction is treated as an autoregressive text generation task, and our proposed UniLP is built on a transformer-based Seq2Seq model, T5. Intuitively, there is no need for negative sampling, and we train UniLP with a standard Seq2Seq objective, i.e., maximizing the likelihood of the target entity sequences with teacher forcing. Specifically, given a KG corpus $\mathcal{G}$, for each fact $(m, s, r, o)$ in $\mathcal{G}$, we verbalize the query $(m, s, r, ?)$ or $(m, ?, r, o)$ across diverse task settings into a unified text prompt to obtain the input sequence $X_i$ according to section 3.3. The corresponding output sequence $Y_i$ are the text mentions of $o$ or $s$, respectively. More detailed input-output pairs refer to the Appendix A.6. In this way, the parameters $\phi$ of the model are optimized in an end-to-end manner by the cross entropy loss, i.e.,

$$\mathcal{L} = arg \max_{\phi} \sum_{i=1}^{|\mathcal{G}|} \log p(Y_i | X_i; \phi). \quad (5)$$

In the inference phase, UniLP directly generates the text of target entity predictions rather than scoring and ranking all possible entities like traditional methods, which can be computationally expensive as the entity set $|\mathcal{E}|$ can be very large. However, the flexible auto-regressive generation may result in generating irrelevant content instead of expected entities. Moreover, the decoding strategies in PLMs strictly select the sequence with the highest conditional probability and are not suitable for link prediction because there

could be multiple valid entities. Therefore, we adopt a Constrained Beam Search strategy [35] to force the model to generate valid entities, i.e., entities in the entity set $\mathcal{V}$. Specifically, we define our constraint in terms of a prefix tree $T$, where each leaf node corresponds to an entity in $\mathcal{V}$, and a path from the root node to a leaf node represents all tokens that constitute a complete mention of the entity. At each step of model generation, the children of current node $t \in T$ represent all succeeding tokens allowed to be generated after the prefix corresponding to the $t$. By using such a prefix-constrained beam search decoding strategy, we obtain top-$K$ entities in the $\mathcal{E}$ without having to score all entities in the KG, where $K$ is the beam size. We assign $-\infty$ for all entities not generated in the decoding stage when calculating the ranking metrics.

## 4 EXPERIMENTS

In this section, we evaluate UniLP on different LP tasks and settings. We conduct extensive experiments to show the effectiveness of our method by answering the following research questions: **RQ1** How does UniLP perform compared to strong baselines across various LP subtasks? **RQ2** How do different key modules in our UniLP framework contribute to the overall performance? **RQ3** How does the context length influence the performance of UniLP? And is UniLP sensitive to context demonstration ordering? **RQ4** How do different decoding strategies influence performance?

### 4.1 Experimental setup

We conduct experiments on 9 LP benchmarks across 4 representative LP subtasks, including static LP, inductive LP, few-shot LP, and temporal LP. The used datasets include (i) *static LP benchmarks*: FB15k-237 [37], WN18RR [10], and Wikidata5M [43]; (ii) *inductive LP benchmarks*: various inductive versions derived from original WN18RR and FB15k-237 by [36]; (iii) *few-shot LP benchmarks*: NELL-One [45] and few-shot version derived from original FB15k-237 by [23]; and (iv) *temporal LP benchmarks*: ICEW14 [13] and ICEWS05-15 [13]. Please refer to the Appendix A.2 for more details on dataset descriptions and statistics. In all our main experiments, we use the T5 base as the backbone of our UniLP. The impact of different T5 parameter sizes is reported in the Appendix A.5. The implementation details and optimal hyperparameters refer to the Appendix A.4.

**Baselines.** To verify the effectiveness of the UniLP framework, we first implement two variants of the model: *UniLP(ent)* and *UniLP(rel)*, adopting the entity-centered demonstrations and the relation-centered demonstrations to model the context information, respectively. Then, we compare against a variety of state-of-the-art baseline methods on LP tasks (Refer to Appendix A.3). The performance of our model is reported on the standard link prediction metrics: Mean Reciprocal Rank (MRR) and Hits@1,3,10. It is worth noting that we have not fine-tuned a basic T5-base model separately as our baseline because KG-S2S is a model that fine-tuning T5-base adaptation to link prediction task using text information, which can be considered as an enhanced version of fine-tuned T5-base.

### 4.2 Performance Comparison with SOTA (RQ1)

**Static LP.** We compare our results with various graph-based and PLM-based methods on the static LP settings. Experimental results are summarized in Table 1. We can observe that (i) UniLP with two

**Table 1: Experiment results of different baseline methods on the static LP datasets. WN18RR and FB15k-237 results are taken from [4]. Wikidata5M results are taken from [30]. The best results are in bold, and the second best results are underlined.**

| Methods | WN18RR | | | | FB15k-237 | | | | Wikidata5M | | | |
|---|---|---|---|---|---|---|---|---|---|---|---|---|
| | MRR | Hits@1 | Hits@3 | Hits@10 | MRR | Hits@1 | Hits@3 | Hits@10 | MRR | Hits@1 | Hits@3 | Hits@10 |
| TransE [1] | .243 | .043 | .441 | .532 | .279 | .198 | .376 | .441 | .253 | .170 | .311 | .392 |
| DistMult [51] | .444 | .412 | .470 | .504 | .281 | .199 | .301 | .446 | 253 | .209 | .278 | .334 |
| ComplEx [38] | .449 | .409 | .469 | .530 | .278 | .194 | .297 | .450 | .308 | .255 | - | .398 |
| ConvE [10] | .456 | .419 | .470 | .531 | .312 | .225 | 341 | .497 | - | - | - | - |
| RotatE [34] | .476 | .428 | .492 | .571 | .338 | .241 | .375 | .533 | .290 | .234 | .322 | .390 |
| CompGCN [39] | .479 | .443 | .494 | .546 | **.355** | .264 | **.390** | **.535** | - | - | - | - |
| KG-BERT [53] | .216 | .041 | .302 | .524 | - | - | - | .420 | - | - | - | - |
| MTL-KGC [16] | .331 | .203 | .383 | .597 | .267 | .172 | .298 | .458 | - | - | - | - |
| StAR [41] | .401 | .243 | .491 | **.709** | .296 | .205 | .322 | 482 | - | - | - | - |
| MLMLM [8] | .502 | .439 | .542 | .611 | - | - | - | - | .223 | .201 | .232 | .264 |
| KEPLER [43] | - | - | - | - | - | - | - | - | .210 | .173 | .224 | .277 |
| GenKGC [44] | - | .287 | .403 | .535 | - | .192 | .355 | .439 | - | - | - | - |
| KGT5 [30] | .508 | .487 | - | .544 | .276 | .210 | - | .414 | .300 | .267 | .318 | .365 |
| KG-S2S [4] | .574 | .531 | .595 | .661 | .336 | .257 | .373 | .498 | - | - | - | - |
| UniLP(ent) | **.588** | **.540** | **.612** | .684 | .344 | **.265** | .378 | 508 | **.373** | **.331** | **.376** | .416 |
| UniLP(rel) | .579 | .536 | .604 | .664 | .331 | .253 | .366 | .496 | .366 | .327 | .359 | **.418** |

**Table 2: Hits@10 metric values of link prediction for KGs in the inductive setting. Results of baselines are taken from [20].**

| Methods | WN18RR | | | | FB15k-237 | | | |
|---|---|---|---|---|---|---|---|---|
| | v1 | v2 | v3 | v4 | v1 | v2 | v3 | v4 |
| NeuralLP [52] | .744 | .689 | .462 | .671 | .529 | .589 | .529 | .559 |
| DRUM [29] | .744 | .689 | .462 | .671 | .529 | .587 | .529 | .559 |
| RuleN [25] | .809 | .782 | .534 | .716 | .498 | .778 | .877 | .856 |
| GraIL [36] | .825 | .787 | .584 | .734 | .642 | .818 | .828 | .893 |
| CoMPILE [24] | .836 | .798 | .607 | .755 | .676 | .830 | .847 | .874 |
| TACT [5] | .840 | .816 | .680 | .766 | .658 | .836 | .852 | .887 |
| SNRI [49] | .872 | .831 | .673 | .833 | .718 | .865 | .896 | .894 |
| ConGLR [20] | .856 | .929 | .704 | .929 | .683 | .860 | .886 | .893 |
| UniLP(ent) | **.973** | **.976** | **.888** | **.969** | **.744** | **.916** | **.937** | **.940** |
| UniLP(rel) | .957 | .955 | .871 | .943 | .714 | .880 | .916 | .922 |

different demonstration templates, i.e., UniLP(ent) and UniLP(rel), consistently outperforms baselines, indicating that both two context demonstration templates in UniLP are capable of capturing context information of query triples and employ them for inference. (ii) For previous SOTA models, the structure-based methods achieve better results compared to part of fully fine-tuned PLMs-based approaches, which demonstrates the topological information of the KG is important when performing link prediction tasks. Moreover, compared to the remaining datasets, structure-based methods perform better on FB15k-237, which can be attributed to semantically meaningless Cartesian Product Relations in FB15k-237. (iii) For the PLMs-based methods, generative methods outperform encoder-only-based methods by a large margin. This indicates decoding generation is more effective than leveraging score functions to rank,

which suffers from unstable negative sampling. (iv) UniLP outperforms KG-S2S, showing that injecting the network signals into the PLMs is effective. That is, by introducing context demonstrations and topology-aware soft prompts, UniLP effectively couples topological and textual information in a contextualized way.

**Inductive LP.** Table 2 reports the experimental results in the inductive setting. We find that both variants of UniLP obtain the best results compared with existing baselines. In particular, UniLP(ent) exceeds the SOTA method by a large margin on all inductive versions of WN18RR (Avg. .952 **vs.** .855) and FB15k-237 (Avg. .884 **vs.** .843). This is consistent with our expectation because PLMs store a large amount of knowledge that enables inherent inductive reasoning ability, and UniLP can effectively harness and unleash the ability of PLMs.

**Few-shot LP.** For a fair comparison, we conduct evaluation under a zero-shot setting like previous PLMs-based methods while the remaining baselines are under a 5-shot setting. Table 3 reports the experimental results on the NELL-one and FB15k-237 datasets. From the results, we can observe that: (i) Even under the zero-shot setting, the PLMs-based methods still outperform most of the previous graph approaches using meta-learning, suggesting that the knowledge acquired by the language model in the pre-training phase enables the model to more robust in low-resource scenarios. (ii) Compared with two representative PLMs-based methods, UniLP obtains a better performance, comparable to the SOTA method that specializes in few-shot settings. That is, UniLP can better utilize PLMs to enable knowledge transfer from the pre-training phase to the unseen ones.

**Temporal LP.** Finally, we verify the ability of UniLP in temporal setting on ICEWS14 and ICEWS05-15 datasets. The results in Table 4 suggest that (i) The improved results of two variants of

**Table 3: Experimental results on the few-shot settings.** † denotes the model exploits the PLMs and is evaluated in the zero-shot setting. Results of baseline are obtained from [23].

| Methods | NELL-One | | | | FB15k-237 | | | |
|---|---|---|---|---|---|---|---|---|
| | MRR | H1 | H5 | H10 | MRR | H1 | H5 | H10 |
| TransE [1] | .168 | .082 | .186 | .345 | .307 | .198 | .419 | .537 |
| TransH [2] | .279 | .162 | .317 | .434 | .284 | .181 | .397 | .503 |
| DistMult [51] | .214 | .140 | .246 | .319 | .237 | .164 | .287 | .378 |
| ComplEx [38] | .239 | .176 | .253 | .364 | .238 | .169 | .281 | .370 |
| GMatching [45] | .176 | .113 | .233 | .294 | .304 | .221 | .410 | .456 |
| MetaR [6] | .261 | .168 | .350 | .437 | .403 | .279 | .551 | .647 |
| FSRL [54] | .153 | .073 | .212 | .319 | .365 | 271 | .456 | .553 |
| FAAN [32] | .284 | .194 | .373 | .451 | .425 | .340 | .459 | .518 |
| GANA [26] | **.344** | **.246** | **.437** | **.517** | **.458** | **.349** | .575 | **.656** |
| StAR† [41] | .260 | .170 | 350 | .450 | - | - | - | - |
| KG-S2S† [4] | .310 | .220 | .410 | .490 | - | - | - | - |
| UniLP(ent) † | .337 | .238 | .430 | .503 | .456 | .346 | **.581** | .654 |

| Methods | ICEWS14 | | | | ICEWS05-15 | | | |
|---|---|---|---|---|---|---|---|---|
| | MRR | H1 | H3 | H10 | MRR | H1 | H3 | H10 |
| TTranse [18] | .255 | .074 | - | .601 | .271 | .084 | - | .616 |
| HyTE [9] | .297 | .108 | .416 | .655 | .316 | .116 | .445 | .681 |
| ATiSE [46] | .550 | .436 | .629 | .750 | .519 | .378 | .606 | .794 |
| DE-SimplE [14] | .526 | .418 | .592 | .725 | .513 | .392 | .578 | .748 |
| Tero [47] | .562 | .468 | .621 | .732 | .586 | .469 | .668 | **.795** |
| TComplEx [17] | .560 | .470 | .610 | .730 | .580 | .490 | .640 | .760 |
| TNTComplEx [17] | .560 | .460 | .610 | .740 | .600 | .500 | .650 | .780 |
| T+TransE [15] | .553 | .437 | .627 | .765 | - | - | - | - |
| T+SimplE [15] | .539 | .439 | .594 | .730 | - | - | - | - |
| KG-S2S [4] | .595 | .516 | .642 | .737 | - | - | - | - |
| UniLP(ent) | **.626** | **.548** | **.671** | **.772** | **.631** | **.550** | **.682** | .789 |
| UniLP(rel) | .614 | .537 | .663 | .748 | .616 | .541 | .664 | .752 |

**Table 4: Experimental results on the temporal setting. The results of baseline are obtained from [15].**

UniLP compared to baselines demonstrate that both local context and global semantics are beneficial for temporal forecasting. (ii) Compared to KG-S2S, UniLP(ent) improves the MRR by 3.1% and Hits@10 3.5% on ICEWS14, which demonstrates the effectiveness of the UniLP for historical context modeling. (iii) Compared to the static LP setting, UniLP obtains more significant improvements. The reason is that temporal datasets focus on specific domains and the temporal dependencies that manifest as structured context can be effectively captured by UniLP.

Overall, by comparing across various task settings and datasets, UniLP consistently and significantly outperforms most specialized methods, which demonstrates the effectiveness and feasibility of UniLP for universally modeling all subtasks in LP. For two different demonstration strategies, UniLP(ent) consistently performs better than UniLP(rel) across various datasets and settings. The potential reason is that, limited by the input length constraint of the language model, the explicit local structure is more beneficial for link prediction compared to implicit global relation semantics. In the following ablation and further analysis experiments, we adopt the

best performing format UniLP(ent) as our backbone, and UniLP refers to Uni(ent) when not otherwise specified.

## 4.3 Ablation Study (RQ2)

In this section, we present ablation studies to support our design. To better understand the contribution of each component in UniLP, We conduct a series of ablation experiments in static and temporal settings, respectively. KG-S2S serves as an ablated variant with only Seq2Seq dropout. The results and meanings of various variants are presented in Table 5. The results reveal that nearly all forms of information are essential because their absence has a detrimental effect on performance. Specifically, we argue that context demonstrations have captured the structure information surrounding the query entity, contributing to link prediction. This can be demonstrated by comparing experimental results between KGS2S and CM + SD, as well as TSP + SD and UniLP. Similarly, by comparing the results between CM + SD and UniLP, as well as TSP + SD and KG-S2S, we also verify that the topology-aware soft prompts can alleviate the structural losses when linearizing the KG structures into text form. Moreover, removing Seq2Seq dropout leads to performance drops, which also validates its necessity and can alleviate potential over-fitting risk.

| | CM | TSP | SD | WN18RR | | ICEWS14 | |
|---|---|---|---|---|---|---|---|
| | | | | MRR | Hits@1 | MRR | Hits@1 |
| Baseline | ✔ | - | - | .582 | .537 | .621 | .544 |
| | - | ✔ | - | .580 | .531 | .616 | .539 |
| | ✔ | - | ✔ | .584 | .538 | .622 | .545 |
| | - | ✔ | ✔ | .581 | .531 | .617 | .539 |
| | ✔ | ✔ | - | **.588** | .539 | .624 | .546 |
| KG-S2S | - | - | ✔ | .575 | 522 | .595 | .516 |
| UniLP | ✔ | ✔ | ✔ | **.588** | **.540** | **.626** | **.548** |

**Table 5: Ablation for the UniLP in static and temporal settings. CM denotes context modeling. TSP denotes topology-aware soft prompts. SD denotes Seq2Seq dropout.**

## 4.4 Analysis on Context Demonstration (RQ3)

**Quantities of Context Demonstrations**. To evaluate the impact of the demonstration amounts provided in the input, we conduct a set of experiments using varying demonstration quantities. Our results, as shown in Figure 4, generally indicate a consistent improvement in performance as the context length increases. This demonstrates that the model performs better as sufficient local structural information is presented. When the context length is relatively short, the model performance fluctuates, which is attributed to the possible introduction of noisy contexts. This observation complies with the nature of graph-based methods that models are susceptible to the sparse issue.

**Order of Context Demonstrations**. We randomly and uniformly sample neighbors as demonstrations in all subtask settings except in the temporal setting, where we order the sampled demonstrations by time. To explore whether UniLP is sensitive to the order

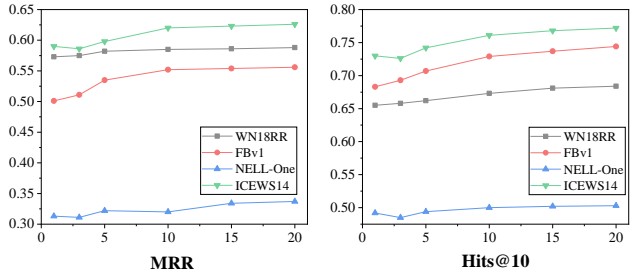

Figure 4: Performances on varying demonstration quantities. FBv1 denotes the FB15k-237 v1 dataset.

of demonstrations, we shuffle the demonstrations to see how the UniLP is affected by the corruption of sequential information. As shown in Figure 5, we find that UniLP is order-insensitive except in the temporal setting. In the temporal setting, the corruption of time order can lead to a deterioration in performance. This observation demonstrates that UniLP has the capability to model temporal dependencies and comprehend the sequential order of demonstrations.

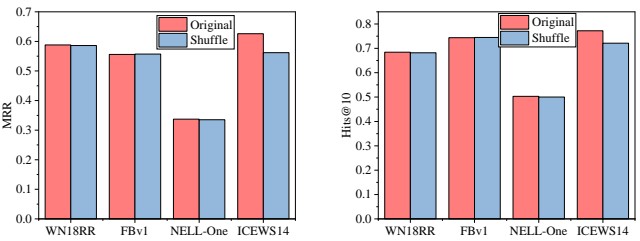

Figure 5: Performance with and without shuffling.

## 4.5    Analysis on Decoding Strategy (RQ4)

**Decoding Strategy**. The results of link prediction have several answers, which motivate us to look into the beam search rather than relying on decoding sequences strictly with the highest conditional probability. We investigate the impact of different decoding strategies in Figure 6 (c). By comparing random sampling search, diverse beam search, beam search, and constrained beam search (UniLP adopts), we find that constrained beam search and standard beam search are always better than the other two. Besides, the constrained beam search further improves the performance of UniLP compared to standard beam search. The reason is that constrained beam search can reduce the search space when decoding by using prefix constraints.

**Influence of Beam Size**. We also explore the impact of beam size in entity generation on performance. From Figure 6 (a), we observe that the MRR performance of UniLP improves as the beam size increases. The reason is that in the decoding process, beam size determines the number of generations for each query, and a smaller beam size implies a smaller search space. With the smaller beam size, it is hard to generate diverse valid entities for queries. Moreover, we find that the model performance changes slightly

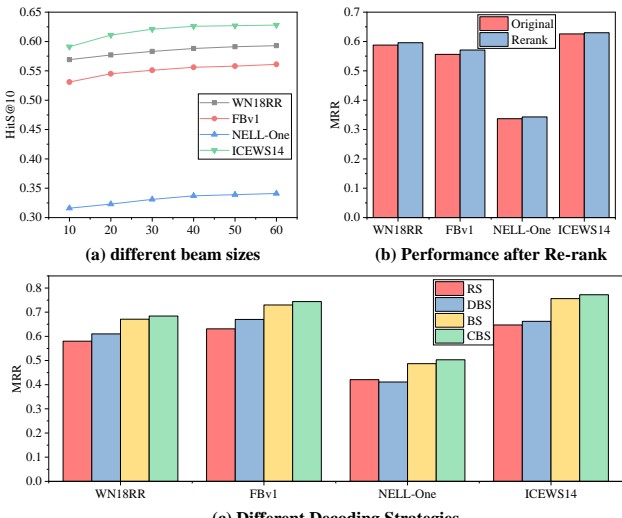

Figure 6: Analysis on decoding strategy.

when beam size $b \geq 40$. Since the inference time goes linearly with the beam size, we finally choose beam size 40 in our experiments.

**Generation Results Re-ranking**. After generating entity candidates with beam search, we find that although UniLP occasionally does not generate ground truth entity with the highest probability, the ground truth entity is always within candidates. To verify this observation, We utilize the most common structure-based method TransE to re-rank the entity candidates. The result is reported in the Figure 6 (b). We find that after re-ranking the entity candidates, UniLP consistently obtains improvements on various task settings, which demonstrates the generation quality of UniLP, and it can inspire future work to devise dedicated knowledge calibration methods to further improve the model performance.

## 5    CONCLUSION

In this paper, we propose a novel prompt-based approach for link prediction, UniLP, which unifies all subtasks under a single Seq2Seq generative framework. To alleviate the information loss caused by conversion of structure to text, we introduce demonstration templates and topology-aware soft prompts to couple topology and text information in a contextualized manner. Our extensive experiments have demonstrated that UniLP outperforms competitive baseline models in various settings. Additionally, we provide an in-depth analysis of the contribution of each component in UniLP. Our work represents a step forward in the unified modeling of link prediction subtasks, and we hope it will inspire future research in this direction. In the future, we plan to extend UniLP to other knowledge-intensive tasks, such as generative recommendation, and information retrieve.

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

# A  APPENDIX

## A.1  Subtask Description

The link prediction tasks across various knowledge graph (KG) structures exhibit significant diversity, rendering model architectures incompatible across these multiple tasks without careful adaptation. These tasks can be categorized into four distinct settings based on the KG structure and downstream task objectives:

- *Static Link Prediction* aims to infer missing entities in static KGs where all entites and relations are observed during training. This is the most common and traditional setting.
- *Inductive Link Prediction* is closer to real-life scenarios, allowing unseen entities during testing. Models in this setting should be designed to focus on entity-independent information in KGs to make up for the gap between testing and training [20].
- *Temporal Link Prediction* predicts a temporally conditioned missing entity in the future given a query quadruple and previous KG snapshots.
- *Few-shot Link Prediction* aims to mitigate the long-tail issue in KGs, predicting facts of a relation with few associated samples.

## A.2  Datasets Statistics

For static LP, we adopt commonly used WN18RR [10], FB15k-237 [37], and Wikidata5M [30]. And we use temporal LP datasets releases from [13] and few-shot LP datasets from [23]. Detailed statistics of all these datasets are shown in Table 6. For inductive LP, we show dataset statistic in Table 7, and due to the space limitation, we abbreviate the name for each dataset. We follow the original split in our experiments.

| Dataset | $|\mathcal{V}|$ | $|\mathcal{R}|$ | $|Train|$ | $|Valid|$ | $|Test|$ |
|---|---|---|---|---|---|
| **Static LP** | | | | | |
| WN18RR | 40,943 | 11 | 86,835 | 3,034 | 3,134 |
| FB15k-237 | 14,541 | 237 | 272,115 | 17,535 | 20,466 |
| Wikidata5M | 4,594,485 | 822 | 20,614,279 | 5,163 | 5,133 |
| **Few-shot LP** | | | | | |
| NELL-One | 68,545 | 358 | 51 | 5 | 11 |
| FB15k-237 | 14,541 | 231 | 75 | 11 | 33 |
| **Temporal LP** | | | | | |
| ICEWS14 | 6,869 | 230 | 72,826 | 8,941 | 8,963 |
| ICEWS05-15 | 68,544 | 358 | 189,635 | 1,004 | 2,158 |

**Table 6: Statistics of the static, few-shot, and temporal datasets.**

## A.3  Baselines

For static LP, we adopt three types of baselines: (i) graph structure-based methods, including TransE [1], DistMult [51], ComplEx [38], ConvE [10], RotatE [34] and CompGCN [39]. (ii) PLMs-based methods (encoder-only), including KG-BERT [53], StAR [41], MLMLM

| | Source KG | | | Target KG | | | |
|---|---|---|---|---|---|---|---|
| | $|\mathcal{R}|$ | $|\mathcal{V}|$ | $|\mathcal{G}|$ | $|\mathcal{R}|$ | $|\mathcal{V}|$ | $|\mathcal{G}|$ | $|Test|$ |
| W1 | 9 | 2,746 | 6,678 | 8 | 922 | 1,618 | 188 |
| W2 | 10 | 6,954 | 18,968 | 10 | 2,757 | 4,011 | 441 |
| W3 | 11 | 12,078 | 32,150 | 11 | 5,084 | 6,327 | 605 |
| W4 | 9 | 3,861 | 9,842 | 9 | 7,084 | 12,334 | 1,429 |
| F1 | 180 | 1,594 | 5,226 | 142 | 1,093 | 1,993 | 205 |
| F2 | 200 | 2,608 | 12,085 | 172 | 1,660 | 4,145 | 478 |
| F3 | 215 | 3,668 | 22,394 | 183 | 2,501 | 7,406 | 865 |
| F4 | 219 | 4,707 | 33,916 | 200 | 3,501 | 11,714 | 1,424 |

**Table 7: Statistics of various inductive versions of WN18RR (W) and FB15k-237 (F).**

[8], and KEPLER [43]. (iii) Generative PLMs-based methods, including GenKGC [44], KGT5 [30], and KG-S2S [4]. For the few-shot LP, we adopt baselines derived from [23] and [6] respectively, including meta-learning based methods and PLMs-based methods. For the inductive LP, we also use two types of baselines, including rule-based baselines and structure-based baselines derived from [36]. For temporal LP, we compare the baselines from [15].

## A.4  Implementation Details

We adopt T5 default settings in our experiments during model training. In terms of hyperparameters, we set the maximum number of epochs as 30-100, depending on the dataset size. we select the batch size from {32, 64, 128}, learning rate from {$5e-3, 1e-3, 5e-4, 1e-4$}, Seq2Seq dropout rate from {0.0, 0.1, 0.2, 0.3, 0.4}. And we use early stopping and model selection on the valid set. The optimal model hyperparameters for each dataset and more details are shown in Table 8. For model evaluation, our model generates the raw text, and we apply regular expressions to remove the special tokens and descriptive spans, leaving only the entity name strings as the final model predictions. We follow the filtered setting proposed in [1] for fair comparison. For each triple in test set, we rank all entities for the query under the object prediction setting and subject prediction setting, and report final mean results. Model is selected by MRR value on valid set.

| | batch size | learning rate | dropout | eval length |
|---|---|---|---|---|
| WN18RR | 64 | 1e-3 | 0.1 | 30 |
| FB15k-237 | 32 | 1e-3 | 0.3 | 30 |
| NELL-One | 128 | 1e-5 | 0.0 | 30 |
| ICEWS14 | 32 | 5e-4 | 0.1 | 30 |
| ICEWS05-15 | 32 | 5e-4 | 0.1 | 30 |

**Table 8: Optimal hyperparameters. Same settings as static LP are adopt under inductive and few-shot versions of FB15k-237 and WN18RR.**

## A.5  Impact of PLMs scalability

We evaluate the performance of our models under different PLMs scalabilities. We report the results in Table 9. We compare the UniLP

with T5-base and T5-small under static LP. From the results, we find that our performance increases with the model size, conforms to the expected trend of better performance with larger models.

| Model | Size | WN18RR | | | | FB15k-237 | | | |
|-------|------|--------|-----|-----|------|-----------|-----|-----|------|
| | | MRR | H1 | H3 | H10 | MRR | H1 | H3 | H10 |
| UniLP(base) | 220M | .588 | .540 | .612 | .684 | .344 | .265 | .378 | .508 |
| UniLP(small) | 60M | .550 | .501 | .583 | .643 | .300 | .236 | .329 | .433 |

**Table 9: Comparison of model performance under different PLMs scalability.**

## A.6 Demonstration Template

**Query:** *<Catherine Ashton, Make a visit, ?, 2014-11-09>*

**Demonstrations:** *<Catherine Ashton, Express intent to meet or negotiate, Mohammad Javad Zarif, 2014-11-04>, «Catherine Ashton, Consult, John Kerry, 2014-11-05>, ..., <Catherine Ashton, Make statement, Iran, 2014-11-06>, ...*

**Input:** *<extra-id-0> Given entity-related context: 2014-11-04 | Express intent to meet or negotiate | Mohammad Javad Zarif [SEP] 2014-11-05 | Consult | John Kerry [SEP] ..., 2014-11-06 | Make statement | Iran [SEP], answer 2014-11-09 | Catherine Asthon | Make a visit | <Mask> <extra-id-1>*

**Output:** *<extra-id-0> Oman <extra-id-1>*

