# OpenReview forum: "UniLP: Unified Topology-aware Generative Framework for Link Prediction in Knowledge Graph"
_ACM.org/TheWebConf/2024/Conference — TheWebConf24_

### Official Review · Reviewer_NvY1 · 2023-11-11

**Novelty:** 6
**Technical Quality:** 6

**Review:**

This paper focuses on Link prediction (LP) in knowledge graph (KG), and aims to unify all subtasks in LP with generative PLM. To this end, the authors solve three challenges, including unified input forms, task-specific context modeling, and topological information encoding. With a context demonstration template and  topology-aware soft prompts, the proposed method UniLP achieves substantial performance gain and provides a real unified end-to-end solution for the whole LP subtasks.

Strengths:

1. The LP task and unified perspective are meaningful and interesting.

2. The motivation and corresponding components in the proposed method are clear and well-motivated.

3. The experiments are solid on diverse LP subtasks, and source code is available.

Weaknesses:

Some method details and experimental settings are unclear.
See details in Questions.

**Questions:**

1. What are the differences between Static LP and Inductive LP?

2. In line 285, edge seems to be more appropriate to indicate the relation between head and tail entities instead of quadruple.

3. What is the advantage of the proposed method over previous methods that train additional GNNs to encode structure information? As the paper claims, the relational perspective seems only fuse the neighbor relations. I think its structural information is weak.

4. Relation-domain and relation-range represent relational semantics of two directions. Why use "domain" and "range"?

5. Further explanation is needed on how to use T5-base in this paper (line 570-574).

6. What is the performance of UniLP on few-shot setting instead of zero-shot in Table 3?

**Ethics Review Description:**

No ethics problem

**Reviewer Confidence:**

3: The reviewer is confident but not certain that the evaluation is correct

**Scope:**

4: The work is relevant to the Web and to the track, and is of broad interest to the community

---

### Official Review · Reviewer_acRz · 2023-11-19

**Novelty:** 4
**Technical Quality:** 5

**Review:**

The paper proposed a unified approach following the seq2seq generative paradigm to tackle the link prediction (LP) tasks facing different situations, such as different application scenarios and graph constructions, instead of handling them independently. The approach introduces context demonstration templates and efficiently fuses textual and topological information through topology-aware soft prompts. Through extensive and comprehensive experiments, the superior performance has been demonstrated even facing various types of LP tasks.

Pros:
1. The paper is clear and well organized. It’s easy to read and follow.
2. Sufficient experiments. The paper performs experiments on a wide range of LP tasks and adopts a variety of SOTA baselines to fully demonstrate the effectiveness of UniLP.

Cons:
1. Missing citation. The paper introduced the construction of context demonstration templates and topology-aware soft prompts based on the seq2seq generative framework, which is proposed by KG-S2S [4]. The citation is missed in Section 3.1. In the “4.3 Ablation Study” section, as we can see, although the demonstration and topology-aware soft prompts has brought some profits, but the main benefits are provided by the seq2seq generative framework.
2. Lack novelty. Given that KG-S2S [4] has integrated static, few-shot, and temporal knowledge graph completion into the seq2seq generative paradigm, the unification of static, inductive, few-shot, and temporal LP tasks seems less innovative, which is the main contribution of this paper.

**Questions:**

1. There is a difference of the KG-S2S’s [4] results in WN18RR between Table 1 and Table 5. Is this due to data errors or is there a different experimental setup?
2. The topology-aware soft prompts are impressive. However, in the “4.3 Ablation Study” section, it seems that the benefits brought by TSP component are modest. And in the FB15k-237 dataset, CompGCN [39] achieved better results. Are there other experiments show that how much structural information can be uncovered by the component?

**Reviewer Confidence:**

2: The reviewer is willing to defend the evaluation, but it is likely that the reviewer did not understand parts of the paper

**Scope:**

3: The work is somewhat relevant to the Web and to the track, and is of narrow interest to a sub-community

---

### Official Review · Reviewer_1s1D · 2023-11-20

**Novelty:** 4
**Technical Quality:** 4

**Review:**

In this paper, the authors introduces a prompt-based learning method for link prediction called UniLP, and experiments demonstrate significant success across various tasks.

Strengths:
1. The author proposes a unified method capable of addressing diverse knowledge graph tasks.
2. The author devises a soft prompt that incorporates topological information within the knowledge graph.
3. The author conducts extensive experiments.

Weakness:
1.  The author claims their primary contribution is the establishment of a UniLP model. However I don't believe building a model like UniLP is crucial:
(1)The author claims their motivation is to unify various subtasks within link prediction. However, in my view, these aren't truly subtasks. Generally, subtasks involve performing different tasks on the same input. For instance, in NLP, a sentence could be used for sentiment analysis or entity extraction. In the four tasks summarized by the author, meta-learning and temporal graph link prediction are distinct tasks or different problem.
(2)  The author needs to explicitly establish the benefits of creating a unified model. I believe it's quite normal to use different models for different problems (different data structures and scenarios). Therefore, a unified UniLP might not be that crucial. Unless, in a real-world scenario, we sometimes need Static LP, sometimes Inductive, at times few-shot, and occasionally Temporal LP, and UniLP can address all these issues simultaneously. Alternatively, similar to prompt learning, using prompts to unify various tasks could achieve few-shot learning or zero-shot learning.
(3) Designing a model that performs well across various scenarios is good, yet the author should assess this model from multiple angles, such as ease of deployment and model complexity, rather than solely focusing on its effectiveness.
2. The author's approach lacks of novelty. The proposed "Entity-centered Demonstrations" essentially concatenate the one-hop neighboring nodes from the graph into the input. Regarding the "Topology-aware soft prompt," despite the authors claiming one of its advantages is the avoidance of training a GCN encoder, their operation is akin to a meaning pooling GCN in practice, albeit this GCN aggregates relationships.
3. UniLP achieved remarkable improvements over traditional methods, but compared to KG-S2S, the enhancement wasn't substantial. This suggests that the modules designed by the author yielded only marginal improvements, while the primary advantage of UniLP might stem from the seq2seq model structure of KG-S2S.
4. The author do not compare the training and inference efficiency of UniLP against traditional methods (e.g., TranSE), which I believe is also crucial.

**Questions:**

1. Why do the authors translate the Link prediction to seq2seq form rather than directly predict links by learned embedding?

**Reviewer Confidence:**

3: The reviewer is confident but not certain that the evaluation is correct

**Scope:**

4: The work is relevant to the Web and to the track, and is of broad interest to the community

---

### Official Review · Reviewer_wbf7 · 2023-11-25

**Novelty:** 4
**Technical Quality:** 4

**Review:**

Work aims to present a unified (ensemble) framework for various types of link prediction (static, temporal, few-shot). The key argument in this paper is: The isolated architectures and chaotic situations make it significant to construct a unified model that can handle multiple link prediction subtasks simultaneously. Authors base it as hypothesis and proposed a unified architecture, which has topology aware prompting, that is induced in a language model, to get the target entity text.

Paper is evaluated in various benchmark, and results are Okay. I say “Okay” because there is no unified state-of-the-art achieved performance agnostic of benchmarks.

The work is nicely written and I appreciate the authors for putting clarity in the work.

However, there are several key topics that are handwaved in the work. The idea of topology aware prompt is not explained. What is the definition of topology aware soft-prompting. Why its needed? Does the prompt really capture a topology? Normally topology of KGs can be used to learn representations and then map to a single data point. Does authors calculate any topology metric such as use of persistent homology? For instance, persistent homology studies the topological features such as components in 0-dimension (e.g., a node), holes in 1-dimension (e.g., a void area bounded by triangle edges) and so on, spread over a scale. Thus, one need not choose a scale beforehand. Can authors empirically prove that soft prompt generated captures topological features? If not, I miss a concrete argument why the work can be seen as “topologically aware”.
In topological idea for KG, one could argue that the theory of persistent homology (PH) to summarize the structures in the graphs in the form of the persistence diagram (PD). Such summarizing is obtained by a process known as filtration. I do not see any such formal process has been followed to design soft prompts which are claimed as topological aware. I am not sure if a prompt generated by combining multiple entity context count as topological aware.
Also adding context while predicting link using PLM is not new and focus on soft-prompting in my humble opinion, does not significantly contribute as scope of full paper.

**Questions:**

Please see above.

**Reviewer Confidence:**

4: The reviewer is certain that the evaluation is correct and very familiar with the relevant literature

**Scope:**

3: The work is somewhat relevant to the Web and to the track, and is of narrow interest to a sub-community

---

### Decision · Program_Chairs · 2024-01-22

**Decision:**

Accept

**Comment:**

In this paper, the authors present UniLP, a prompt-based learning approach designed for unified link prediction tasks. UniLP follows the seq2seq generative paradigm, addressing various link prediction challenges across different scenarios and graph constructions, rather than treating them as separate entities. The authors tackle three key challenges: unifying input forms, modeling task-specific context, and encoding topological information. Through a context demonstration template and topology-aware soft prompts, the UniLP method significantly improves performance and offers a comprehensive, unified solution for all link prediction subtasks.

 The paper fits well into the scope of the Semantics track. The originality of the presented approach is acknowledged by the majority of the reviewers. Also the technical quality of the paper ist considered adequate, especially after the complements the authors have provided during the rebuttal.

 Pros:
 1. The LP task and unified perspective presented in the paper are meaningful and interesting.
 2. The paper presents a soft prompt that incorporates topological information within the knowledge graph.
 3. The paper performs experiments on a wide range of LP tasks and adopts a variety of state of the art baselines
 4. The source code is available

 Cons:
 1. The idea of topology aware prompt is not explained, but the authors provide information in the rebuttal
 2. The need for a unified approach is not fully motivated.
 3. Some reviewers consider the novelty of the presented approach as limited.
 4. UniLP achieved remarkable improvements over traditional methods, but compared to KG-S2S, the enhancement wasn't substantial, suggesting that the primary advantage of UniLP might stem from the seq2seq model structure of KG-S2S.
 5. The evaluation of the original paper could be improved. However, the authors provided complementary information and results in the rebuttal.